Extending density surface models to include multiple and double-observer survey data

http://orcid.org/0000-0002-9640-6755 Miller David L. 1 dave@ninepointeightone.net
http://orcid.org/0000-0001-5433-4733 Fifield David 2
Wakefield Ewan 3
Sigourney Douglas B. 4
1 Centre for Research into Ecological and Environmental Modelling and School of Mathematics and Statistics, University of St Andrews , St Andrews , Scotland
2 Wildlife Research Division, Science and Technology Branch, Environment and Climate Change Canada , Mount Pearl, NL , Canada
3 Institute of Biodiversity Animal Health and Comparative Medicine, University of Glasgow , Glasgow , Scotland
4 Integrated Statistics , Woods Hole, MA , United States of America
Shang Yilun
Electronic publication date: 2021 Sep 2
Publication date: 2021
Volume: 9
Electronic Location ID: e12113
Received 2021 Apr 23; Accepted 2021 Aug 14
Copyright year: 2021
License: This is an open access article, free of all copyright, made available under the Creative Commons Public Domain Dedication. This work may be freely reproduced, distributed, transmitted, modified, built upon, or otherwise used by anyone for any lawful purpose.
License URL: https://creativecommons.org/publicdomain/zero/1.0/

Keywords: Density surface model, Distance sampling, Generalized additive model, Spatial modelling, Variance propagation, Abundance estimation

Funding: U.S. Navy’s Living Marine Resources program N39430-17-C-1982 National Marine Fisheries Service M14PG00005 US Department of the Interior, Bureau of Ocean Energy Management, Environmental Studies Program, Washington, DC NEC-16-011-01-FY18 UK Natural Environmental Research Council (NERC) NE/M017990/1 David L. Miller was funded by OPNAV N45 and the SURTASS LFA Settlement Agreement, being managed by the U.S. Navy’s Living Marine Resources program under Contract No. N39430-17-C-1982, collaboration between Douglas B. Sigourney and David L. Miller was also facilitated by the DenMod working group (https://synergy.st-andrews.ac.uk/denmod/) funded under the same agreement. The survey that the fin whale data originate from was funded through two inter-agency agreements with the National Marine Fisheries Service: inter-agency agreement number M14PG00005 with the US Department of the Interior, Bureau of Ocean Energy Management, Environmental Studies Program, Washington, DC and inter-agency agreement number NEC-16-011-01-FY18 with the US Navy. The survey that the fulmar data originate from was funded by the UK Natural Environmental Research Council (NERC) grant NE/M017990/1. The funders had no role in study design, data collection and analysis, decision to publish, or preparation of the manuscript.

==============================
Spatial models of density and abundance are widely used in both ecological research (e.g., to study habitat use) and wildlife management (e.g., for population monitoring and environmental impact assessment). Increasingly, modellers are tasked with integrating data from multiple sources, collected via different observation processes. Distance sampling is an efficient and widely used survey and analysis technique. Within this framework, observation processes are modelled via detection functions. We seek to take multiple data sources and fit them in a single spatial model. Density surface models (DSMs) are a two-stage approach: first accounting for detectability via distance sampling methods, then modelling distribution via a generalized additive model. However, current software and theory does not address the issue of multiple data sources. We extend the DSM approach to accommodate data from multiple surveys, collected via conventional distance sampling, double-observer distance sampling (used to account for incomplete detection at zero distance) and strip transects. Variance propagation ensures that uncertainty is correctly accounted for in final estimates of abundance. Methods described here are implemented in the dsm R package. We briefly analyse two datasets to illustrate these new developments. Our new methodology enables data from multiple distance sampling surveys of different types to be treated in a single spatial model, enabling more robust abundance estimation, potentially over wider geographical or temporal domains.

Introduction

As ecological data are amassed over time, the job of the modeller becomes increasingly difficult. Faced with a large number of potentially useful datasets from multiple surveys with different field protocols, the question becomes “how can I include all this information?” Methods to address this kind of question should be clear in their asssumptions and implications as well as having easy-to-use software implementations using methodological frameworks that researchers are familiar with. In this paper we attempt to address this problem for the case of spatially-explicit abundance estimation from distance sampling data.

Spatially-explicit estimates of abundance are used for a variety of purposes in conservation and ecological settings. Distance sampling-based techniques (Buckland et al., 2001) are extremely popular ways of estimating abundance or density of biological populations. As techniques have advanced, methods have been developed to incorporate spatial information (e.g., environmental covariates) (Hedley & Buckland, 2004; Johnson, Laake & Ver Hoef, 2010; Yuan et al., 2017), moving towards model-based, spatially-explicit abundance estimates. One approach is density surface modelling (DSMs; Hedley & Buckland, 2004; Miller et al., 2013), which combine detectability information using standard distance sampling methods with a spatial model using the generalized additive modelling framework (Wood, 2017). DSMs have been used to obtain abundance estimates for populations where the individuals are not uniformly distributed over the study area (Harihar, Pandav & MacMillan, 2014), to inform spatial planning in impact assessments (Winiarski et al., 2014, 2013) and to mitigate negative impacts of military operations (Roberts et al., 2016).

Currently DSMs are most often applied to data from a single survey with a single detection function, sometimes using one or more covariates to model variation in detectability (multiple covariate distance sampling, MCDS; Marques & Buckland (2004)). Here we extend these models to the case where we have detection functions that account for missing observations at zero distance (mark-recapture distance sampling, MRDS; Burt et al. (2014)) and where it is necessary or desirable to integrate data from multiple surveys (hereafter, platforms) into one model. This could simply be because each available dataset is limited in its spatial or temporal extent. Modellers may also find it preferable to include data in a single spatial model rather than attempting model averaging over several smaller models.

Essentially we wish to combine different observation processes, modelled in different ways via one spatial model. Integrating these different observation processes is currently possible via related fully Bayesian approaches (e.g., Sigourney et al., 2020), though these may require considerable time investment in terms of both understanding and fitting the model.

We envisage several possible situations where this kind of approach will be useful (though we note this list is non-exhaustive):Combining data from multiple distance sampling surveys. These may have used different platforms, different observers or other factors which change the form of the detection process. For example, Roberts et al. (2016) combined 23 years of cetacean observations, some surveys were aerial and others ship-based; altitudes, observer positions and weather/sea conditions differed between surveys, leading to a number of detection models.

A single survey where different field methods are used for different behaviours of the animal. For example seabirds-at-sea (SAS) protocols (e.g., Tasker et al., 1984; Camphuysen et al., 2004), where distances are recorded to birds detected on the water, whereas birds detected in flight are recorded using a strip transect methodology (assuming detection is certain out to some distance). Another example is the case where distances were usually recorded, but when observers were overwhelmed in high density areas, they switched into a strip transect mode to ensure all animals were recorded (Clarke et al., 2020).

When both point and line transect surveys have been used to survey a species and these need to be combined into one spatial analysis. For example combining camera trap distance sampling (Howe et al., 2017) with conventional line transect surveys.

Combining mark-recapture distance sampling and conventional distance sampling survey data.

In this article, we show how data can be integrated from the above situations by combining multiple observation models (detection functions) to build one spatially-explicit abundance model that accounts for varying detectability conditions.

Materials and Methods

Density surface modelling

Density surface models (DSMs) are a two-stage approach where the first stage of the model is to estimate the observation process and then in the second stage uses the estimated detectability in a spatial model of abundance. The observation process is modelled using a detection function (Buckland et al., 2001), which is estimated using the distances from the line or point transect to the detections. The detection function models the decrease in probability of detection with increasing distance, potentially including the effects of covariates such as sea state or weather. Once the detection process is modelled, the detectability can be estimated unconditional of distance from the observer (by integrating distance out of the detection function; Buckland et al. (2001)).

The detectability of objects subsequently contributes to an offset in a spatial model of the counts. For line transects, counts are aggregated in small line sections called segments (Miller et al., 2013); for point transects, counts are aggregated to the points (which we also refer to as segments for brevity). The counts are assumed to come from some count distribution and are modelled (on the link scale) as a sum of flexible smooth functions of environmental and spatial covariates (such as location, sea surface temperature, bathymetry, etc) as part of a generalized additive model (GAM; Wood (2017)). The linear predictor for the model includes an offset term consisting of the product of the area of the segment and the detectability in that segment (this product can be thought of as an “effective area”, analogous to the “effective strip width” or “effective radius” used in line/point transect distance sampling). The DSM can be written as:

(1) E[ni|β,λ,p(θ^;zi)]=aip(θ^;zi)exp⁡(β0+∑mfm(xim)),

where ni, the number of individuals in segment i (of area ai), follows some flexible count distribution such as Tweedie or negative binomial (where above we assume a log link). p(θ^;zi) is the detection probability for objects in segment i, with detection covariates zi which vary by segment (we refer to aip(θ^;zi) as the offset). The fm are smooth functions of environmental covariates, xim, represented by a basis expansion (i.e., fm(x)=∑jβjbj(x) for some basis functions bj) penalized by a (sum of) quadratic penalty (or penalties); β0 is an intercept term, included in parameter vector β; λ is a vector of smoothing parameters which control the wiggliness of the smooth components of the model (Wood, 2017). We may also include unpenalized terms in the model, as we would in a generalized linear model.

In practice, these models can be fitted using the R packages Distance (Miller et al., 2019) for simple detection function fitting and dsm (Miller et al., 2013) for spatial modelling. dsm uses the mgcv package to fit GAMs using restricted maximum likelihood (REML).

One might be tempted to extend the above model to multi-platform data by using one detection function with a categorical covariate controlling for differences in platform (e.g., a factor for boat/plane). This approach makes the assumption that the same detection function form was used for all data (e.g., that all data originated from a detection process that was hazard-rate in shape), which is unlikely to be realistic when there are different platforms (e.g., when combining aerial and shipboard surveys). Moreover, such an approach does not address when one of the surveys is a strip transect with detection assumed to be perfect, nor when there are multiple MRDS detection functions present.

Multi-detection function density surface modelling

We wish to build models that include data from multiple platforms. Platforms include: physically different surveys conducted via different means (e.g., aerial and shipboard, or surveys taking place at different times), different survey protocols taking place in the same survey (e.g., in SAS, birds on the water via line transect vs. those flying via strip transect) or some combination of these.

Each platform has a different detection function, which we index by k = 1, …, K. We index segments associated to a given platform (k) by jk = 1, …, nk. For each platform we have a different corresponding probability of detection, derived from that detection function’s parameters (θk) and covariates (zjk). We then write the detectability as pk(θk; zjk) for an observation in segment j using detection function k. For strip/plot transects, we assume perfect detection, so pk(θk; zjk) = 1. Once we have fitted detection functions and estimated pk(θ^k;zjk), we can then fit model (1) as:

(2) E[njk|β,λ,pk(θ^k;zjk)]=ajkpk(θ^k;zjk)exp⁡(β0+∑mfm(xm,jk)),

where notation is as in (1) with the addition that xm, jk indicates the value of covariate m in segment j for platform k.

In the case where different survey protocols are used simultaneously (e.g., during a seabird survey from a ship, when birds on the water are recorded using distance sampling and birds in flight are recorded simultaneously using plot sampling), we treat each as a separate platform. In this case, the segment data are duplicated, so we have one copy for each platform (e.g., in the previous scenario, with n1 segments we analyse 2n1 segments in total, n1 for birds on the water and n1 for flying birds). Examples of doing this in practice are given in Supplementary Material B.

Differing density by platform/observation type

Including multiple platforms as in (2) assumes that any differences in observed counts are a result of detectability alone and that mean density does not differ between the two platforms. This may be unrealistic, especially when surveys were conducted from different platforms at different times (e.g., seasonal surveys) or when platforms operate simultaneously but record animals in differ behavioural states (e.g., SAS). A simple adaptation to model (2) would be to include a per-platform intercept, βk:

(3) E[njk|β,λ,pk(θ^k;zjk)]=ajkpk(θ^k;zjk)exp⁡(βk+∑mfm(xm,jk)).

Model (3) assumes that the density only shifts the intercept via βk and has no effect on the fms, which may be overly restrictive. We can extend our model using the hierarchical GAM framework (Pedersen et al., 2019) to allow the smooth parts of the model to vary by platform (factor-smooth interactions). We allow information to be shared between these smooths, such as how wiggly they are (shared smoothing parameters) or that they have similar shapes (smoothing towards a global term). To do so, we extend (2) as:

(4) E[njk|β,λ,pk(θ^k;zjk)]=ajkpk(θ^k;zjk)exp⁡(β0+∑m1fm1(xm1,jk)+∑m2fm2(xm2,jk,k)),

where the m1 smooths are as in (3) and m2 are factor-smooth interactions. Using a spatial smooth (f(x, y)) as an example, we can then fit a spatial smooth for each platform: we have f(x, y, k) for k = 1,…, K. We may choose a subset of terms in the model that seem most likely to vary by platform or, include all terms as factor-smooths. Pedersen et al. (2019) enumerate all the models possible under this framework.

Incomplete detection at zero distance and availability

A fundamental assumption of distance sampling is that objects at zero distances are observed with certainty: that is, g(0) = 1 if g is the detection function (Buckland et al., 2001). In one approach to dealing with a violation of this assumption, two observers (or teams of observers) can be used to set-up a capture-recapture experiment where the probability of observing an animal at zero distance is estimated by considering one observer as setting-up trials for the other when animals are detected (mark-recapture distance sampling; MRDS). Using this approach we estimate g(0; zg(0), θg(0)) where θg(0) are parameters specifically for the estimation of g(0) and zg(0) are (optional) covariates. Burt et al. (2014) give details of models for g(0; θg(0)). Here we consider only the independent observer mode of MRDS where each observer’s detections are trials for the other.

Including MRDS models into the DSM framework is simply a case of using g(0; θg(0)) as an additional multiplier on the detection probability in the offset. We re-write p(θ^;zi) in model (1) as the product g(0;θg(0))p(θ^;zi). Accordingly we can index the MRDS model with k and include it as one of our multiple surveys in models (2), (3) or (4).

As we will see below, we can also account for animals not being available to be detected (e.g., due to diving behaviour) in our models by using an estimate from other sources (such as tag data) and including this in the same way as g(0). For now we assume that the estimate of availability is independent of the surveys (i.e., data were collected at a different time, using different methods, perhaps in a different area) and therefore that we can add the squared coefficient of variation to the DSM’s to obtain a total uncertainty on our abundance estimates. Availability estimates can be specified at the segment level, so can vary between platform (see below).

Strip and plot sampling

If we assume that all objects within a given distance of the sampler are detected with certainty, we have strip or plot transects (as analogues to line and point transects). Strip transects are common when video/photo surveys are conducted from planes or drones, so detectability is not an issue or, when the truncation distance is sufficiently small that it is believed that all objects will be detected. In this case we simply replace the p(θ;zi) in model (1) with one.

Variance estimation

Each fitted detection function included in the DSM has its own covariance matrix for that model’s parameters. If we assume independence between the detection functions, we can create a joint covariance matrix as a block diagonal matrix with each block representing one of the K platforms: Vθ = diag(Vθ1, Vθ2, …, VθK), where Vθk is the covariance matrix for the parameters of the detection function for platform k. When a strip/plot transect is used there is no corresponding θk and therefore no uncertainty, so it does not appear in Vθ.

If we are willing to assume independence between the detection function and spatial model, we could apply the delta method (Seber, 1987) to combine the detection function and GAM variances. The delta method is unappealing if there are implicitly spatial covariates in the detection model (e.g., wind speed varies in space) as there will be non-zero covariance between those model components. Instead we can adapt the variance propagation approach of Bravington, Miller & Hedley (2021) to include multiple detection functions. This approach uses a quadratic approximation to the detection function in a refit of the spatial model to adjust the detectability estimates in light of the additional spatial information from the GAM covariates. The quadratic adjustment takes the form of a random effect with mean zero and covariance matrix Vθ, so the detection function uncertainty is included in the final variance estimate. In this way Vθ can be plugged-in to the variance propagation method of Bravington, Miller & Hedley, 2021 and a posterior covariance matrix for all model components (Vβ, θ) can then be estimated. Note that the assumption of independence is made a priori but the variance propagation procedure can estimate the off-diagonal elements of Vβ, θ. In the next section we show how this uncertainty about the model parameters can be applied to the uncertainty in abundance estimates.

Estimating abundance and its variance

DSMs are most often used to provide estimates of abundance over the study area (or some subset(s) of it), which are calculated as sums of predictions over grids (Miller et al., 2013). We can express this in matrix form as N^=aexp⁡Xpβ, where a is a row vector of grid cell areas, Xp is the matrix that maps model coefficients to predictions (a design matrix for the predictions) and β^ are the estimated GAM parameters.

To calculate the uncertainty in estimates of N^, we can use posterior sampling (sometimes referred to as parametric bootstrapping), using the procedure outlined in (Wood, 2017, Section 7.2.6). We use the posterior distribution of β to generate possible abundance estimates, then calculate appropriate summary statistics. The following algorithm can be used:

1. Calculate the matrix Xp using the model and prediction grid.

2. For b ∈ {1, …, B}:Generate new model parameters βb∗.

Calculate the new linear predictor η∗=Xpβb∗ for each prediction cell.

Calculate predictions on the response scale N^b∗=aexp⁡Xpβb∗.

Store N^b∗ for this iteration.

3. Calculate empirical variance and mean of the B stored N^b∗ values.

The posterior distribution of β is approximately multivariate normal with mean β and variance Vβ, θ. To sample from this distribution we can directly sample using a multivariate normal random number generator, but may obtain better results by avoiding this assumption and using the Metropolis–Hastings algorithm to sample from the model posterior (such a sampler is provided as part of the package mgcv, see Supplementary Material B for an example of its use).

This method is extremely general. For example, we can extend this algorithm to calculate per-cell estimates by calculating per-cell abundances at 2.(c) above. A more complex example is when we include βk in the model as in (3). In this case we are assuming that although the study area is the same, the densities are different for the different platforms. When making predictions we predict for each platform (k = 1, …, K) then sum these per-platform predictions.

Software implementation

The above methods are implemented in the R package dsm version 2.3.1 (Miller et al., 2021), available on CRAN. Detection functions can be fitted using the R packages Distance version 1.0.2 (Miller et al., 2019) or mrds version 2.2.4 (Laake et al., 2020). The software implementation puts some requirements on the data. Specifically, one must be able to identify each observation as being from a particular platform and therefore detection function. The same must be true for the segments. Examples of data setup are available as part of Supplementary Material B.

Example data

We give two brief examples illustrating the above method. In the first example we combined two different platforms. The second example explores the use of models (3) and (4). We have included the code to run these examples in Supplementary Material B and data is available at the following DOI 10.5281/zenodo.5116140.

Multiple surveys with uncertain detection on the trackline-fin whales

These data consist of observations of fin whales (Balaenoptera physalus) as part of NOAA’s Atlantic Marine Assessment Program for Protected Species. Data were collected during two distance sampling surveys: one shipboard (requiring adjustment for g(0)) and one aerial (requiring an adjustment for availability and g(0)). The left panel of Fig. 1 shows effort and detections. Details on field methods are available in Sigourney et al. (2020). Here we reproduce the analysis from that article using the DSM approach.

Figure 1 Fin whale data, predictions and uncertainty.

Left: transects (lines, colour-coded by platform) and detections (points, size scaled to observed group sizes) of fin whales; inset shows position of the study area off the Atlantic coast of North America. Middle: predictions from the fin whale model, density is calculated as animals per km2. Right: corresponding map of coefficient of variation for the predictions. Comparable maps from the fully Bayesian model of Sigourney et al. (2020) are available at https://doi.org/10.7717/peerj.8226/fig-3; breaks are as in those figures for comparability.

Shipboard observations (truncated at 6 km) were conducted in independent observer mode mark-recapture distance sampling with the model for g(0) only including distance as a covariate; a hazard-rate detection function was used, with Beaufort wind speed and a subjective measure of the effect of weather conditions on detectability as covariates. Observations from the aerial survey (truncated at 900 m) were modelled using a hazard-rate detection function with Beaufort wind speed as a covariate; a fixed g(0) and availability adjustment was applied based on results in Palka et al. (2017).

We specified the following DSM:

(5) E(ni)=aipig(0)iuiexp[f(DIST125)+f(DEPTH)+f(DIST2SHORE)+f(SST)]

where for segment i, ni is the count (assumed to be Tweedie distributed), ai is the area, ui is the availability (one for shipboard segments, 0.37 for aerial segments), pi is the probability of detection and g(0)i is the correction for detection at zero distance (0.67 for aerial segments and based on the MRDS model for shipboard segments). f indicates a smooth constructed using thin plate regression splines with shrinkage (Marra & Wood, 2011), each with a maximum basis size of five. The covariates used were DIST125, distance to the 125 m isobath; DEPTH, depth; DIST2SHORE, distance to shore; SST, sea surface temperature. We used the variance propagation method of Bravington, Miller & Hedley, 2021 to propagate uncertainty from the two detection functions. Model checking gave reasonable results (see Supplementary Material A).

Abundance was calculated on a grid over shelf and shelf slope waters of the North East United States and South East Canada from Delaware to Nova Scotia. To estimate uncertainty in abundance, we can first take a Metropolis–Hastings sample from the posterior of our model (post variance propagation). The resulting samples capture uncertainty in the spatial model, aerial detection function, shipboard detection function and shipboard g(0). Assuming independence between the aerial estimates of g(0) and availability, we can use the fact that squared CVs add to combine uncertainty for these estimates in a way which is comparable to the estimate given in Sigourney et al. (2020).

Multiple platforms on one survey-fulmars

RRS Discovery conducted a survey for seabirds in the mid-Atlantic in June 2017 as part of cruise DY080. Figure 2 show the transects and observations (and Fig. S1 shows the study area in context of the North Atlantic). A modified version of the Eastern Canada Seabirds At Sea (ECSAS) protocol (Gjerdrum, Fifield & Wilhelm, 2012) was used while on effort, comprising a line transect survey for birds on the water and a strip transect survey for birds in flight. A single observer, located on one side of the bridge (varied according to conditions to avoid, e.g., glare) searched for all birds flying or sitting on the water within a 300 m wide strip to one side of the transect line, while a second person recorded bird sightings and sighting conditions. Birds were detected using the naked eye and identified using 10 × 40 or 8 × 40 binoculars. Birds on the water were detected in one of four distance bins: [0–50 m], (50–100 m], (100–200 m] and (200–300 m]. A “snapshot” method was used to record birds first detected in flight, flagging records of these birds if they were within a 300 m × 300 m box at the moment of the snapshot (Tasker et al., 1984). We selected observations of fulmars (Fulmarus glacialis) for analysis. Due to their distinctive plumage and flight behaviour, fulmars are generally easy to distinguish from other species at sea, but confusion can occur with large immature gulls. However, the latter are virtually absent from the study area during summer (Wakefield et al., 2021), so misidentification of fulmars was assumed to be negligible.

Figure 2 Fulmar observations and effort.

Study area (grey outline box, as in Fig. S1) for the fulmar data with effort (solid line) and faceted by platform (detection type: flying or swimming). Dots show locations of observations. See Fig. S1 for the study area in context.

We therefore have two platforms. For fulmars on the water, we fitted a half-normal detection function, with precipitation (factor; yes/no) and visibility (in km) as covariates. For flying birds, we simply assume that detection is perfect out to 300 m (the width of the strip transect).

To explore the formulations defined above, we fitted models based on (2)–(4). Explanatory covariates were limited to spatial smooths of projected location (x, y) with variations on how the differing animal behaviour (“platform” in our terminology) was accounted for:(A) Based on (2), with linear predictor β0 + f(x, y), the bivariate smooth of space used thin plate regression splines with shrinkage (Marra & Wood, 2011).

(B) Based on (3), with linear predictor βk + f(x, y) where βk is an intercept depending on the behaviour. The spatial smooth was as above.

(C) Based on (4), with linear predictor β0 + f(x, y, k) where we have a factor-smooth interaction for the spatial effect. The factor-smooth model creates a smooth for each platform as deviations from a reference level.

All models were constructed so that the maximum basis size was 100. As above, the variance for the detection function was propagated using the method of Bravington, Miller & Hedley, 2021. A complete analysis of these data is available in Wakefield et al. (2021).

We compared our models using the following techniques:Residual model checking procedures outlined in e.g., (Wood, 2017, Chapter 7) can be used to assess the models and potentially remove from consideration models that have violated assumptions.

AIC scores can be used to compare models, as we might usually do for GAMs. Since here we use the same detection model in each case we need not include the detection function AIC in our considerations (but if different detection functions were used we would).

Goodness-of-fit can be assessed in terms of the number of observed vs. predicted animals swimming or flying. Aggregating at the level of some binned covariate is important as the smoother will (by its nature) tend to predict small values where exact zeros were observed. We can aggregate by any covariate (whether or not it is included in the model), here we use the platform covariate to assess our fitted models since we are trying to address differences in platform.

For models that include additional terms to account for platform (models B and C), plotting the difference surface between per-platform predictions spatially. We expect that if that extra complexity is not needed we would not see a difference in predicted densities.

Results

Multiple surveys with uncertain detection on the trackline: fin whales

Checks of fit indicated reasonable conformity to model assumptions (see Supplementary Material B) and effective degrees of freedom for each term in the fitted model were well below the specified maximum of five. We obtained an abundance of 3,935 fin whales. Combining the uncertainty from the posterior sample (spatial model, aerial detection function, shipboard detection function, and shipboard g(0)) with aerial estimates of g(0) and availability gave a total CV of 0.42. Middle and right panels of Fig. 1 shows maps of predictions and coefficient of variation (without the fixed parameter additions just discussed) over the prediction grid cells, patterns there match those in the original analysis.

Multiple platforms on one survey: fulmars

Residual model checking showed slightly better conformity to assumptions for the factor-smooth model (see Supplementary Material B). This model performed best, followed by the factor model (ΔAIC = 11.5) and then the model that assumes no differences in density (ΔAIC = 32). Table 1 shows that the model using one additional factor covariate for platform performs best, with predicted values closest to those observed. Figure 3 shows the differences between platforms for the two models; in this case there are differences between the predictions. These differences are larger (less white in the plot) for the factor-smooth model but the factor model also shows differences between the two platforms.

Figure 3 Prediction differences within fulmar models.

Differences between per-platform density predictions (flying minus swimming) for the factor and factor-smooth models for the fulmar data. Figure S3 shows the separate surfaces for each platform from which these plots are derived. In both cases we see greater numbers of fulmars predicted swimming in the north of the study region, especially around 31° W, with no differences south of 46° N. We can see much larger differences (in both directions) in the factor-smooth model.

Table 1 Observed vs. expected diagnostics for the fulmar models.

	Observed	No factor	Factor	Factor-smooth	
Swimming	501	405	501	490	
Flying	533	632	516	517	
χ2 statistic		38.26	0.457	0.704	
Note:

Observed vs. expected numbers of fulmars aggregated by behaviour (swimming/flying) for each model. χ2 statistics are given in the final row as a summary comparison. From these results, it appears that the factor model gives the best match to the data at this aggregation.

Differences between the models’ predictions are not reflected very well in the spatial predictions for each model shown in Fig. 4, top row, as the per-platform effect has been averaged out. Differences are better seen with Fig. 3 and Table 1, which show evidence there are differences in density between the two platforms. Figure 3 shows a key difference between the two models which account for differences in platform: the factor-smooth interaction is more flexible and can fit better to the observed data, meaning that differences between the two platforms’ predictions can be greater, whereas the factor only model must make a compromise in fit between the two platforms (up to the change in level allowed by the factor platform covariate).

Figure 4 Comparison of fulmar model predicitons and uncertainty.

Comparison of density predictions (top row) and coefficients of variation (bottom row) by model (columns) for the fulmar data. Predictions and uncertainty appear to be very similar between models when aggregated but see also Fig. S3.

Uncertainty estimates are lower with the factor-smooth model: the lower uncertainty areas (CV ≤ 0.4) remain roughly the same, but the higher uncertainty area has decreased. Figure S3 shows the per-platform plots and highlights where the uncertainty originates: the platform for flying birds has much higher uncertainty than that for swimming birds, despite there being more observations of flying birds (383 swimming vs. 460 flying) and the spatial patterns of observations being very similar (Fig. 2). Biologically, the additional uncertainty in the data from birds in flight may be down to transient weather conditions not fully captured in the spatial smooth (whereas variation in the birds on the water may be better described by the spatial smooth). Supplementary Material A and Fig. S2 shows a map of sighting conditions.

Discussion

Many articles conclude with the phrase “further data are needed” and while this may be true, exactly how to utilise these new data while integrating previous information can be tricky. The issue becomes serious when estimating the abundance of endangered populations, when we must ensure that data are used wisely to inform management policy. If we are able to combine data from multiple sources we can potentially make better inferences about populations and make better decisions about how to ensure their conservation.

In this article we have addressed the common issue of combining distance sampling data from multiple platforms with different observation processes, into one spatial model. Our approach shares information between the platforms at the spatial modelling stage while also ensuring that variance is propagated correctly between the observation and spatial models. Compared to fitting multiple spatial models and averaging the results, our approach allows for information to be shared via the spatial model (using the variance propagation approch of Bravington, Miller & Hedley, 2021), not via post hoc averaging (it also avoids the thorny issue of deciding how to weight such an average). The extension using factor-smooth interactions allows the spatial model to vary between platforms if necessary, while reducing to a combined model where appropriate. From a practical perspective, model construction is more efficient for the modeller: we only need to fit one model and the factor-smooth construction allows for the sharing of information between data sources, so if there are not sufficient observations for a given platform, the model can “borrow strength” from the others. Having fitted the model, we are still able to make predictions at the platform level to disentangle the full model and explore our results.

Though we have used two marine examples here, we wish to stress that these methods are applicable in any distance sampling (or combination of distance and plot/strip transect sampling) situation. For example combining point and line transect methods may be particularly useful for terrestrial surveys of birds where both methods can be used, or (as mentioned above) when combining camera trap distance sampling with line transects.

A perennial problem with advocating new methodology is in demarcating where it is appropriate to use that new method. From our experience we recommend the use of the factor-smooth approach (given in (4)) as a starting point and simplifying if there is no evidence that the additional complexity is warranted (this can be investigated by plotting predictions at the segment-level against each other). The process we outlined for model selection gives a starting point, but of course usual model checking for both detection functions and generalized additive models should be followed.

Sigourney et al. (2020) fitted a hierarchical Bayesian model to the fin whale data which included informative priors on availability process, aerial g(0) and group size (their model was fitted to number of observed groups rather than number of individuals), making an exact comparison with our method difficult. Their model gave N^ = 4,012 with CV = 0.32 (our estimate was 3,935, CV = 0.42). Presumably the inclusion of informative priors helps the hierarchical model lower its uncertainty. Numerical results and plots show that our new method is comparable to a more complex hierarchical Bayesian analysis, though clearly there are some areas for improvement.

Our fulmar example takes data which have been collected using the de facto standard methodology for surveying seabirds at sea: simultaneously recording birds in flight via plot sampling and birds on the water via distance sampling. Previous techniques for deriving abundance and distribution estimates from such data implicitly ignored potential differences in the distributions of birds in these two behavioural states. The example above suggests that although the distribution of fulmars in flight and on the water was broadly correlated, the ratio of birds in these two states also varied in space. Our framework allows for the modelling these effects giving more accurate abundance estimates as well as potentially provide biological insights and allow the generation of hypotheses regarding the behaviour of the animals in different conditions.

There are two main limitations to the models presented here. The first, is that there is a restriction on the kinds of covariates we can model in the detection function: we can only fit the “count model” as defined in Miller et al. (2013) (count per segment as response, detectability entering the offset) and so detection function covariates may only vary at the segment level, not at the level of the individual observation. To circumvent this limitation, we can use factor-smooth interactions where the factor levels are binned versions of the covariates and duplicate the segments for each of these levels. For example, binning group sizes to be small/medium/large, then fitting a smooth to each of these levels. Further details of this approach are given in Bravington, Miller & Hedley (2021). The second deficiency is that we cannot share detection parameters between platforms, as each detection function is fitted separately. So, for example if multiple surveys are carried out from the same vessel at different times, we cannot use a single “vessel” covariate which is jointly estimated between the models. It is possible to do this, but it would require a more complex fitting framework than we use here. Such situations can be handled relatively easily in software such as JAGS (Plummer, 2003) or Nimble (de Valpine et al., 2017), in which case mgcv::jagam (Wood, 2016) can be used with a dsm model to setup smoothers in a fully Bayesian framework. For given common situations, further work could make this process semi-automatic (in the sense that JAGS/Nimble code could be automatically generated to link parameters), making the construction of these more complex models easier.

Extension to further detection functions is a clear next step, including those for g(0) which incorporate other observer configurations and accounting for point independence (Buckland, Laake & Borchers, 2010). An additional extension is the use of other observation processes beyond detection functions. For example models to account for animals that are not always available to be observed, such as diving seabirds or cetaceans (e.g., Borchers et al., 2013). Provided that such an availability bias correction enters the model via the offset, the Bravington, Miller & Hedley (2021) procedure can be used; this could include models built using GAMs themselves. We hope this article prompts further exploration of these possibilities.

Supplemental Information

Supplemental Information 1 Fulmar study area in context.

Position of the fulmar study area (thick grey box) within the North Atlantic, with bathymetry (coloured lines) and land (grey polygons) for reference. Transects are shown as the thicker black lines, note that only data from those within the study area were used in our analysis.

Click here for additional data file.

Supplemental Information 2 Spatial distribution of weather covariates used in the detection function for the fulmar analysis.

Lines indicate transect positions, thick grey box the study area as previously.

Click here for additional data file.

Supplemental Information 3 Comparison of platforms for the fulmar models.

Predictions (top row) and coefficients of variation (bottom row) by platform and model (columns). The factor model (model (3) is used in the left two columns and factor-smooth model (model (4)) in the right two columns. The black lines show the survey segments. Compared to Fig. 4 we see differences in the predictions and the uncertainty. As the model formulation dictates, platform differences in the factor model dictate the overall level of the plot, whereas the factor-smooth models vary in their pattern too. This is particularly notable for the uncertainty plots.

Click here for additional data file.

DLM wishes to thank Megan Ferguson and Natalie Kelly for useful discussions and testing.

Additional Information and Declarations

Competing Interests

Author Contributions

Data Availability

Douglas B. Sigourney is employed by Integrated Statistics.

David L. Miller conceived and designed the experiments, performed the experiments, analyzed the data, prepared figures and/or tables, authored or reviewed drafts of the paper, and approved the final draft.

David Fifield conceived and designed the experiments, authored or reviewed drafts of the paper, and approved the final draft.

Ewan Wakefield conceived and designed the experiments, performed the experiments, prepared figures and/or tables, authored or reviewed drafts of the paper, contributed the fulmar data, and approved the final draft.

Douglas B Sigourney conceived and designed the experiments, prepared figures and/or tables, authored or reviewed drafts of the paper, contributed the fin whale data, and approved the final draft.

The following information was supplied regarding data availability:

All code and data are available at Zenodo: DL Miller. (2021). dill/multiddf-supp-materials: Code version for revision of the paper. (paper-revision). Zenodo. https://doi.org/10.5281/zenodo.5116140.

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
