# Peer review of "Extending density surface models to include multiple and double-observer survey data"

_PeerJ, doi:10.7717/peerj.12113_

## Round 0.1 · original submission · Major Revisions

We have received two detailed reports on the manuscript. Both reviewers are in general positive about the paper. They have provided detailed suggestions and comments. Please revise the paper and provide a one-to-one response.

·

Basic reporting

Overall the writing was of high quality (see Specific Comments for a few grammar errors and typos I caught), the introduction and literature cited were well referenced, and the figures were great. I was able to access the data and code via David Miller's github site. I suggest this material be archived to a publicly available repository after manuscript acceptance.

Experimental design

I found that the methods, data, and results to be reliable.

Validity of the findings

The statistical methods developed are sound and the text identifies how the research fills into existing knowledge gaps. Code and data provide a useful template from which readers can replicate the author's results and exemplify how they can apply these methods to their own research.

Additional comments

Miller and colleagues describe a framework for modeling animal count data from multiple types of surveys, including strip transects, point transects, and various distance sampling setups. Specifically, they describe how a common spatial model can be assumed for animal density, and how this can be integrated with different types of observation processes. The framework uses a two-stage framework, whereby inference is first made about detection probability, and those results are incorporated as offsets into spatial models. Importantly, uncertainty about detection probability is propagated into abundance estimates via a variance propagation method. They show how this framework can be used by applying it to fin whale and fulmar data sets.

I thought the manuscript was well written and represented competent research. In addition, the software extensions to the \texttt{dsm} R package should allow other researchers to apply these types of analyses to their own data sets. I view the latter as the manuscript's primary strength. I only have a few very minor comments, mostly on grammar, below. However, the authors might want to reflect on whether the fulmar surfaces should be added together instead of averaged.

Note that I am going to attach a properly formatted review; however here is some raw .tex giving specific comments

\item Line 55, for a very recent example of combining various detection methods via integrated likelihood, see Conn et al. 2021 (https://doi.org/10.1371/journal.pone.0251130). Just FYI, I'm not suggesting the authors need to cite this as it uses very similar ideas to the Sigourney paper.

\item Lines 58-62. Various grammar fixes needed here

\item Line 81. Is is worth saying that for line transects it is usually perpendicular distance from the transect line rather than distance when the sampler detects the animal?

\item Lines 150-151. This looks like a subsection heading with a formatting issue?

\item Lines 159-160. Worth talking about this in discussion... e.g. future extensions to software that can partially accommodate detection heterogeneity via point independence

\item Lines 202-206. It would be slightly clearer if things were given subscripts here, maybe with ``For each $b \in \{1,2,\cdots,B \}$:" (not a big deal though)

\item Lines 216-217. The bird example is an interesting one. In this case, wouldn't you add the two (flying and non-flying) rather than take a mean?

\item Lines 237-246. This seems to be the first time availability is mentioned in this paper. Is there capacity in \texttt{dsm} to put in availability and it's associated variance? If so, is it a new or existing feature? It seems like it might be a useful topic to discuss earlier in the paper (e.g. combining platforms with and without availability issues)

\item Line 272. Were there ``unknown species" recorded?

\item Lines 274-275. Grammar

\item Line 277. A bit of a fourth-decimal-place question, but were $x$ and $y$ lat-long or in projected space?

\item Line 286. Wow! I guess this gives the GAMs flexibility to capture extremely fine scale patterns if they exist, but it's quite a bit larger than default \texttt{mgcv} values. Just curious if this is something the authors often do in DSMs?

\item Discussion. Currently there's a paragraph about the fin whale example here but nothing about the fulmar example. Should lessons from the fulmar example be covered here to balance things out?

·

Basic reporting

The manuscript is well written and addresses a potentially very useful extension to current approaches to the estimation of spatial variation in abundance using distance sampling data. I have a few general comments listed below, but these are mostly minor.
The main area for improvement in my opinion, is to provide a bit more general framing of the new methodology in the Introduction and Discussion, to really drive home the usefulness of the approach. I think this is needed to provide a bit of a “hook” to sell the approach to users, and to perhaps to explain how the methodology fits into more general thinking about modelling spatial variation in abundance. Perhaps also framing the approach terms of a more general trend in ecological statistics to use multiple data types in a single model to achieve more than would be possible with each data type in isolation could be the way to approach this. To my mind, there are parallels here with methods such as Integrated Population Models, or with various kinds of Bayesian models that seamlessly integrate different kinds of data and sampling processes into a common framework to furnish strong ecological insights and maximise the efficiency with which all of the available data are used. I realise the approach in this manuscript it not the same as these methods (which are often Bayesian), but I strongly feel that they are scratching the same itch – combining disparate data that have something to say about an ecological question.

Experimental design

I have no criticism of the analytical approaches that are employed in the manuscript. The methodology is clearly presented and likely to be easily applicable to other comparable data sets. The fact that it has already been integrated into the dsm R package will make the approach widely known and easily accessible. Provision of code and data for the examples on the author’s github site will also facilitate adoption of the methods.

Validity of the findings

I have no criticism to make as to the validity of the findings. The work is technically sound and all underlying code and data have both been made available for scrutiny.

Additional comments

1. The examples in the paper are largely framed around marine surveys from ships and planes. I wonder if a more general explanation would help, including stressing how these methods could apply to a multitude of different species and ecosystems. I would well envisage these approaches being applied to estimation of spatial abundance models for animals and plants in other environments.
2. I realise it is widely used in distance sampling literature, but I find the term “platform” a bit confusing. I always think of the “platform” as being the ship, aircraft etc on which the observers travel. This becomes confusing when for example there are multiple data types being collected from the same ship (as is the case for the fulmar example in the text, in which the multiple simultaneous survey types are referred to as “platforms”). I would prefer “survey type” or similar be used to refer to the different data types (CDS, MRDS, plot sampling) while the vehicles can be referred to as “platforms”.
3. Line 61. “leading *to* a number of detection models”.
4. Lines 57 – 72. These are certainly a good examples of situations where this methodology could be readily applied. Perhaps mention that this is a non-exhaustive list of possibilities?
5. Lines 373-376. References to using an alternative approach here with mgcv and jagam will be a bit opaque to non-specialists. Can I suggest an explanation in more general terms of what is intended here would be worthwhile? I love the idea of making this process automatic (or nearly so).
6. Figure 1 could do with a scale bar, and possibly an inset map to illustrate the location – I presume this is somewhere on the Atlantic coast of North America, but this isn’t clear from the figure.
7. Table 1. It’s pretty obvious from these results which model fits best, however the caption says “fit diagnostics”, without actually providing any measure of fit for each model – it is left to the reader to decide which model fits best. Would some Chi-squared statistics in another row of the table make this clearer to the reader at a glance?
8. Figure 3 – what are the units of density? Birds per square kilometer, or something else? Same applies to Figure 4 – no units provided.
9. Figure S1 – the caption doesn’t say what the dashed lines are. Presumably these are the transects? Did data from the transects outside the study area also contribute to estimation of the detection processes? Also, the grey shadings used for the bathymetry contours are very difficult to distinguish. Would a different colour scheme, or means of mapping the bathymetry such as a coloured raster plot work better?
10. Figure S3 – state units of density, as mentioned above.

---

## Round 0.2 · accepted · Accept

Both reviewers agree for acceptance. Congratulations.

·

Basic reporting

No comment

Experimental design

No comment

Validity of the findings

No comment

Additional comments

The authors have done a great job responding to my (minor) comments on the previous draft.

·

Basic reporting

The manuscript has been greatly improved by the revision. Thank you for so thoroughly addressing all of the comments raised by both reviewers. The responses to all of the comments are thoughtful and well integrated into the revised manuscript.

Experimental design

No further comments

Validity of the findings

No further comments

Additional comments

One very minor error is that there are duplicate references to Miller et al. 2019 in the reference list.